# How Constructivist Environment Changes Perception of Learning: Physics Is Fun

Grzegorz P. Karwasz [1,*] and Katarzyna Wyborska [1,2]

1   Didactics of Physics Division, Institute of Physics, Faculty of Physics, Astronomy and Applied Informatics, University Nicolaus Copernicus, 87-100 Toruń, Poland
2   Primary School in Dąbrowa Biskupia, 88-133 Dąbrowa Biskupia, Poland
*   Correspondence: karwasz@fizyka.umk.pl

**Abstract:** The global availability of information makes its selection difficult, but at the same time it allows for the construction of teaching without the particular prior knowledge of students. However, it requires teachers to learn new abilities, such as developing a much broader coverage of the subject, explanations of illy solutions, and knowledge of different ways of thinking and the mental needs of pupils (pedagogical knowledge contents). We show examples of such teaching in physics in several quite different environments: from school classes to workshops for 3–4-year-old children, interactive lectures for children's universities, ad hoc explanations in science museums for secondary school students, to public lectures in didactics at international congresses. Every specific environment requires different approaches, but the contents may remain similar: innovative, constructivist, and interactive approaches assure a successful outcome in any didactical situation.

**Keywords:** learning environments; learning sciences; teacher education; constructivist approach; cognitivism

## 1. Introduction

*Motivation: "Physics Was Not My Favorite Subject"*

In spite of technological progress, contemporary schools face new, demanding requirements; this is not the transmission of knowledge, but rather the forming of complex competencies. They include, first of all, reasoning and the ability to select and evaluate the available information, but also meta-competences, as capacities of organising collaborations, presenting one's own knowledge, etc., i.e., both social and personal competences. This has been precisely worded by Sheer et al. [1]:

> The mandate of schools is to unfold the personality of every student and to build a strong character with a sense of responsibility for democracy and community. This implies developing skills of reflection, interpretation of different information and other complex meta-competences.

Physics played a crucial role in the formation of both ancient and modern science. Four "elements" specified by Empedocles (and recalled by Aristotle in Metaphysics 988a, 26) are still taught as such in today's schools. The nature of time and space, being the main contents of Aristotle's Physics, is still the point of departure for popular science bestsellers, such as The order of time by Carlo Rovelli [2] (and similar books by Roger Penrose and Stephen Hawking [3], Chris Ransford [4], etc.). Galileo, in his discussion on accelerated motion (Dialogo dei Massimi Sistemi [5]) put forward the basis for what we call modern science: a repeatable experiment and its mathematical description (see the English Wikipedia for adjectives assigned to Galileo as a founder of modern science). Newton not only formulated mathematical laws of nature but, as stated by Robert Crease, made the world rational:

> The arrival of the Newtonian universe, was attractive, liberating and even comforting to many of those in the 17th and 18th centuries; it promised that the world

was not the chaotic, confusing and threatening place it seemed to be—ruled by occult powers and full of enigmatic events—but was simple, elegant and intelligible. Newton's work helped human beings to understand in a new way the basic issues that human beings seek: what they could know, how they should act and what they might hope for. [6]

This cultural change was much more important than what we teach traditionally about Newton in school. Not in regard to the impact of modern physics, the "principle of relativity" and "uncertainty" (indeterminazione in Italian) became an integral part of contemporary philosophy. Despite its historical and cultural role (or perhaps because of it), physics in school is not an easy subject, and its traditional teaching does not seem to fit the postulates of Sheer et al. [1]. A common opinion is "physics was not my favorite subject at school". This subjective statement is supported by educational studies. Osborne [7], in England, showed, in a sample of approximately 3.5 thousand students attending middle schools, that as far as their interest in geography, biology, and English remained high (it changed from "like very much" to "like much") between the third and fifth form, it dropped drastically (from "like" to "dislike") in physics. A bigger drop (from "like much" to "dislike") was reported only for chemistry (and for French, it changed between "dislike much" and "dislike"); see Figure 3 in [7]. A study by the European Union (so-called the Rocard Report [8]) warned of the drastic fall (by 50% between 1995 and 2005) of the number of university graduates in the subjects of physics and engineering in France, Germany, and the Netherlands. A slighter fall was seen in the USA, and none was detected in the Republic of Korea.

In Poland, physics is an obligatory subject in elementary and secondary school. In the recent (published in January 2023) study of the Polish Physical Society [9], performed on a sample of 350 scientists and teachers, 73% of the responders postulated changes in teaching physics to make it attractive at university level. A similar percentage confirms a negative image of physics in society. In the same study, as many as 82% of the responders indicated the capacity of analysing facts and drawing conclusions as the main feature of physicists.

The teaching of physics in Poland suffered from the recent (started in 2015 and fully operative only in 2021) reform of the national system. The system had been already changed in 1999, with the perspective of entering Poland into the EU, and resembled mixed Italian/French solutions: 5 years of elementary school (starting at the age of 7), 3 years of middle school (junior high school), and 3 years of Lyceum (senior high school, or vocational school). Physics was taught for three years in middle school and two years in Lyceum. However, the "Law on School Education" act of 2016 re-introduced the system which was in place before the EU adhesion, i.e., 8 years of elementary plus 4 years of Lyceum.

The main shock for teaching physics came from the lack of teachers at the elementary level: textbooks were prepared in a rush without didactical testing and in many schools, teachers of mathematics were swapped over to teaching physics. So far, this recent "reform" still lacks a quantitative evaluation: Figure 1 shows the latest published (2014) national score in the mature exam (the UK A-level equivalent) in physics.

As seen in Figure 1, the results were pitiful in the group that chose physics as an additional (i.e., not main) subject; the distribution of votes is Poisson's-like, meaning a good score (and even the sufficiency) was "a rare event". Teaching physics in Poland is efficient only for a small group of students who are particularly interested in this subject, so they choose an extended form of the exam (Figure 1b). This contrasts with the presumed role of physics as forma mentis, i.e., with its cultural influence in society. Actions must be taken, not to train scientific experts, but to increase the positive image of physics in general.

The reasons for the low "popularity "of physics as a school subject are three-fold. The first is the very structure of physics as a science, which requires necessarily both the theory and the experiment. This is not the case, say, for philosophy or biology: the former rarely appeals to experiments and in the latter, the role of theories (such as "conservation laws") is not as dominant as it is in physics. The second reason is the didactics of physics: teaching requires both humanistic (verbal communication) as well as mathematical capacities. The

problems in physics start from a (verbal) description of the situation, possibly existing in nature, then requires a mathematical formulation, solving the equations, accepting results that are feasible, and, again, the verbal formulation of the answer. The third most important reason for the poor performance of students in physics (particularly in Poland, as we have witnessed) is the traditional, transmission-like way of teaching, with few experiments, almost no multimedia, and lacking any interdisciplinary connections [11,12].

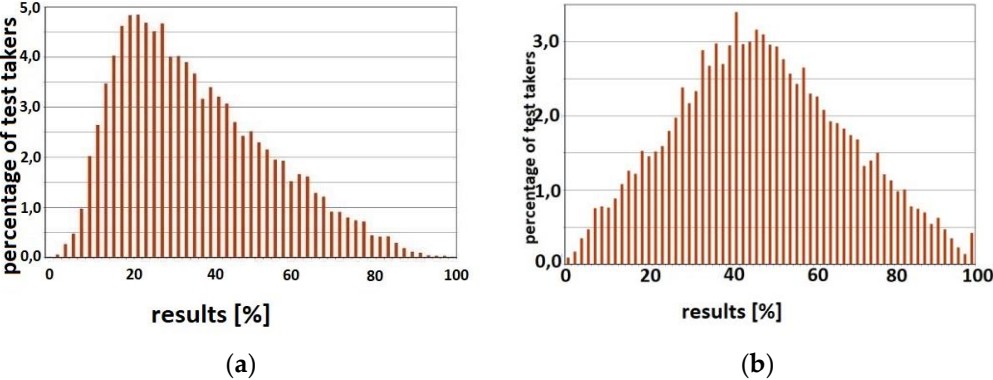

| (a) | (b) |

**Figure 1.** Matura exams in Poland in Physics in 2014. (**a**) Standard level—the results resemble Poisson's distribution: high scores are rare events. The dominant statistic is at 21% of the possible score; (**b**) Augmented level: the results are distributed into a Gaussian, but the majority still stays below the sufficiency level. The substantial difference between the two distributions indicates that the requirements for the maturity exam were incoherent with the notions and competences transmitted in schools. Source: Report of Polish Ministry of Education [10], reprinted with permission.

The difficulties in teaching physics were detected around half a century ago, so physicists, teachers, and scientists undertook a whole range of actions to make their subject more attractive. Already in the 1960s, the first institutions for science divulgation, such as the Exploratorium in San Francisco, were founded [13,14]. These centres gather huge numbers of visitors (such as the "Kopernik Science Centre", which opened in Warsaw in 2010 and was visited by 1 million people in the first year of activity), and physics is the main leading thread there [14]. However, in spite of that, school groups are the main target; the impact on the ways of teaching (and learning) physics is still, in Poland, marginal. Teachers are not prepared to deliver lessons inside the centres, and the staff employed there have no specific competence in didactics.

The second strategy to attempt to make the didactics of physics easier would be new and affordable approaches to teaching. We mention here the master Feynman's Lectures in Physics, the Conceptual Physics by Paul Hewitt with its thirteen editions (also in Poland, under the name "Physics around us"), or the recent series by Leonard Susskind and Art Friedman, for example, their Quantum Physics, published in Italy with the subtitle "The necessary minimum to practice (good) physics" [15]. The common denominator of these books is the step-by-step illustrative and narrative didactics. However, the target group of these books remains, essentially, university students.

The third "line of attack/defence" by physicists are books for the general public that explain big philosophical questions. Apart from the already mentioned writers, we add such authors as Piergiorgio Odifreddi (mathematician) and Antonino Zichichi (experimental nuclear physicist at CERN) [16] in Italy, John Barrow (professor in mathematical physics) and John Tipler (experimental physicist) [17] in the UK, Michał Heller (priest and professor of cosmology) in Poland [18], and many others. Their books have gained popularity thanks to their interdisciplinary (and philosophical) approach.

As an expression of efforts to revitalise the teaching of physics, a global-range network, Groupe International de Recherche sur l'Enseignement de la Physique (GIREP), has been created; its congresses gather hundreds of specialists in didactics each year. GIREP is an important forum for the exchange of innovative ideas. However, the path from the



didactical idea to its successful implementation is not straightforward in practice: high-level scientific concepts of modern physics rarely find a way to enter the school curricula; see, for ex., refs. [19–21].

In the present paper, we review a series of actions undertaken by us over approximately twenty years in order to increase the attractiveness of physics as a school subject (and, in consequence, also the number of university students in this discipline). The research question is: can we translate high-level principles of physics, as (i) an interdisciplinary science, (ii) based on experiments, but (iii) working on abstract concepts, to the cognitive level of children and teenagers? Which forms and contents should be developed to retain the durable interest of the audience/school class/general public? Is this interest profitable in terms of the knowledge of physics, apart from a mere phenomenology? We experimented in many ways, and the answer is complex, but in general, it is "yes!".

## 2. Need for New Approaches

Transferring knowledge in an encyclopaedic way can result in oblivion. The rapidly progressing developments of civilisational require a new approach to teaching to be developed. From observing pupils in school, we realise that memorising rules or definitions is pointless nowadays. Increasing the student's motivation for further work can only be done by asking questions and independently constructing knowledge based on their own experiences, which in turn will lead to understanding the discussed phenomena or physical concepts.

The overarching goal in the teaching process should be to equip students with skills that will allow them to apply their acquired knowledge in new situations in a rapidly developing world. Deepening curiosity for discovering and understanding the surrounding world and developing a willingness to use skills for human needs should take place through interdisciplinary teaching. Depicting physics as a field of science which is inseparable from other subjects will lead to a better understanding of the nature which surrounds us and, consequently, to the more effective use of acquired knowledge in everyday life. Physics is particularly suitable for searching scientific laws in objects that surround us; for example, see our virtual and real exhibitions under the nickname "Physics and Toys" [22]. On average, these pages attract a thousand visitors a day.

Students who can independently construct knowledge during lessons learn faster and are more involved in the teaching–learning process because they ask themselves more questions and pose possible hypotheses. The introduction of changes in the teaching process is inevitable and should be supported by the knowledge and application of the latest trends in pedagogy and general psychology, and, above all, by understanding the teaching and learning processes in the cognitive–constructivist approach.

There are many noteworthy publications on the methodology of teaching physics, but the observation of the present study shows that, quite often, teachers treat the education process as strict in guiding the child's development, replacing the process of the self-learning of the pupil with a mere transmission of notions, which in turn leads to the acquisition of a collection of information, instead of creating conditions for the child to construct their knowledge and use their own life experiences. A manifestation of development is a method for the child's improved functioning in the world, coping with its complexity thanks to their acquired skills and adopted attitudes towards the immediate environment [23].

There is a general agreement that the didactical process should undergo substantial changes. The old, traditional teaching model should be abandoned in favour of a constructivist model, with a variety of methods, means, and approaches. "Students like 'hands-on', 'inquiry-based' learning, laboratory experiments, and learning environments that encouraged independent thinking"; see [24] and the references therein.

*2.1. Hyper-Constructivism*

The term "constructivism" was developed in the 1930s from Jean Piaget's observation of how newly born children, day by day, construct their image of the world. Piaget assumes that Man constantly constructs his knowledge, as result of new experiences, in accordance with the principles of assimilation and accommodation. According to Piaget, children construct their knowledge about the world on their own, as they become integrated into the environment, and their cognitive schemas are their independent constructions resulting from constant exploration. In order to know objects, the subject must have active contact with them.

Piaget's theory was not only ground-breaking for thinking about cognitive and social development, but it also provided inspiration for the theory and practice of education, especially those directions in which the importance of constructing knowledge is emphasised. The foundation of teaching, he believed, was a discovery:

> To understand is to discover, or reconstruct by rediscovery, and such conditions must be complied with if in the future individuals are to be formed who are capable of production and creativity and not simply repetition. [25]

Jerome Bruner, the creator of the theory of mental development, who strongly emphasised the role of social interactions, can be considered a continuator of Piaget's research. Author of original didactic concepts, he paid particular attention to the role and conditions of stimulating the cognitive activity of children and youths and the development of positive motivation. In the 1950s, the term "cognitive science" appeared. Jerome Bruner, with the advent of early computers, joined psychology, cybernetics, and linguistics into a unique science, exploring the view of the external world in the human mind [26].

Lee Shulman is to be considered the third founder of innovative didactics. He introduced the concept of pedagogical content knowledge (PCK) [27], stressing that the range of abilities and competencies of the teacher must go far beyond the mere competencies of the subject he/she teaches. Teachers' masteries must include the psychology, pedagogy, and history of his/her matter.

We add to these competencies a principle called "9:1": the teacher must know not only a single, correct answer, but also nine possible wrong answers and, moreover, must know why the pupil provides a particular illy answer [28]. In other words, the teacher must be able to "dig up" the pupil's way of thinking, in his/her previous knowledge, in the (junk) external information, i.e., the reasons for the didactical failure.

Wide applications of constructivism, cognitivism, and PCK to teaching science dates to the beginning of the XXI century. However, it was only a moderate success; no educational revolution happened. One of the reasons is that the new didactics require much more advanced skills of teachers. He/she must be able to correctly guide the ways of thinking of pupils, ensuring their knowledge is not too tight and not too loose. John Nesbit and Liu Quing stated [29]:

> The effectiveness of inquiry-based learning depends on the guidance provided by teachers. An unguided or minimally guided inquiry may not work for students who have less previous knowledge or ability in the subject area. When the demands of the learning activities exceed students' abilities, their learning is blocked and they may develop misunderstandings about the topic.

Further, the teacher must also accept losing the role of a "guru", i.e., of the expert who never fails. Obviously, in an authoritarian school system, it would erode the position of the teacher. Even in the UK, Paul Newton et al. described it as follows [30]:

> We do have average and below average teachers who actually don't have the skills to run discussion groups. I think that it is quite a high level skill for teachers. [ . . . ] Teachers need to be confident enough to accept that they may not know the answers, this may [ . . . ] discourage some teachers from allowing the situation to occur in the first place.

The eroding of the authority of the teacher may be avoided if we change the approach from ex-cathedral teaching to a constructivist-like dialogue. Then, the teacher needs not show his/her intellectual supremacy but instead become a partner in the common search for the scientific truth. As far as teachers, say, in Poland, declare the will to use the constructivist approach, they miss practical examples. Keith Taber, resuming the current situation with the implementation of new didactics, postulates "re-energising constructivist work by suggesting new perspectives and approaches" [31].

We propose, first of all as a practical application, in different environments, a new approach that combines the three fundaments: constructivism, cognitivism, and PCK. The name we use for this new methodology [28,32–34] is "hyper-constructivism" (H-C). Briefly, it consists of a guided, collective discovery of the law/phenomena/facts, with the aim to form the correct understanding in the mind of every single student. Out of nine science-specific pedagogical content approaches in teaching sciences, which include, among others, a flat transmission of notions, academically rigorous teaching, developing skills, activity-driven orientation, discovery orientation, etc. [35], our methodology is closest to a guided inquiry that "constitutes a community of learners".

The starting point in our methodology is critical thinking, i.e., Cartesian's basis to modern scientific methodology, used also by the OECD (Organisation for Economic Co-operation and Development) in the AHELO (Assessment of Learning Outcomes in Higher Education) evaluation of university systems [36]. So, the unit of teaching (a lesson, a workshop, or an interactive lecture in a great hall) starts by triggering critical thinking. This is independently if we speak to 3–4-year-old children or adult teachers, as noticed by Knigth et al. [37], who found that even in early childhood, collaborative thinking is the key to critical thinking. So, we invite an open discussion, addressing the whole audience: "Why do objects fall?", at the beginning of our lesson on mechanics. It is not important who the audience is. With small children, such a lesson would finish with free playtime with wooden toys (which are easy to bring within carry-on luggage); for selected, advanced students in Korea, the lesson ends with the General Relativity Theory of Einstein (see further in this paper).

Constructivism for us is a practical teaching methodology, not a theory. However, we go beyond Piaget's idea of the individual construction of knowledge and beyond an alternative understanding of this term, as a social process of "negotiating" knowledge [38]. Further, students become the main actors in the process of active discovery [39]. Keeping in mind all these constraints, we practice (and theorise [32–34]) a blended approach, in which the knowledge of students is formed in an interactive process, within the vivid interaction with the group (a school class, audience at public lectures, children at workshops, etc.), but the teacher precisely monitors the cognitive ideas which appear during the lesson/lecture in the minds of the participants. For this reason, we apply the term "hyper-constructivism".

Nowadays, the baggage of notions in the possession of even small children is so vast, we simply must choose the best information, fitting our didactical goal, rather than assisting in the Socrates-like "midwifery" way. Using publicly available sources of knowledge, e.g., on the Internet, we can use these in the joint creation of knowledge in any subject, at any level or in any environment. This (collective) knowledge is the basis of interactive narration in the H-C method. Here, the role of the teacher is extremely important as a wise cicerone, knowing in advance which of the (hidden for pupils) paths leads to erroneous arrival points. The teacher, allowing an open discussion, thanks to his PCK skills, leads (in a delicate way) the thinking of the students, to make them follow (one of the possible) correct paths, and to provide them with the opportunity to perform their own, individual (and almost autonomous) learning.

In the H-C method, the knowledge becomes constructed in a spontaneous way. The teacher is responsible for setting the right path for the students to follow. The teacher is to some extent just an observer, a coordinator, and offers help if the students need it. Constructing knowledge can be based on the collection of information that students

possess, and, if needed, referring to their own sources (textbooks, other printed materials, or the Internet).

The teacher's role is no longer to organise and systematise knowledge but to conduct analytical group reasoning. The point is to choose the one that is the most convincing, logically and scientifically correct from all available cognitive paths leading to the set goal.

The hyper-constructivist lesson is necessarily based on the interaction within the class. No experiment will be shown until the class expresses their opinion on its outcome. It does not matter much whether the answer is right or wrong; we need a working hypothesis to start the process of step-by-step deducing and verifying. As noticed in the case of demonstrations during university lectures in physics [40], if no prediction of the outcome of the experiment had been asked, the level of correct responses is low (0.58), and it increases to 0.8–0.85 if the students made some predictions (even if they were wrong).

Our approach, which appeals to the pre-existing knowledge of the audience, is, by force, interdisciplinary. We search in the minds of pupils for any relation/remembrance/connection to phenomena which could be useful in constructing knowledge, and this is the goal of the lesson. This process stimulates a brainstorm in the group, which is an important methodological competence on one hand, and on the other hand, rectifies the knowledge of pupils. Again, it is not important if the construction takes place in an elementary school or with an audience of university students. Moreover, the younger the pupils are, the more "strange" the ideas that they have are and the more flexible they are in an open discussion. The goal is to "clarify concepts and enable to link scientific ideas with other ideas" [41].

The next essential point in hyper-constructivist teaching is the need for a step-by-step consensus. As we apply group reasoning, every member of the group must be (pretty) convinced that he/she con-divides the outcome for the reasoning; otherwise, she/he would be alienated from the next steps. Newton et al. [30] summarised it as follows:

> It is not enough for students just to hear explanations from experts (e.g., teachers, books, films, computers); they also need to practice using the ideas for themselves. 'The' answers to 'the' questions need to become 'their' answers to 'their' questions.

*2.2. Neo-Realism: Objects and Multimedia*

The hyper-constructivist method, as previously mentioned, is based on the knowledge already possessed by students/pupils. However, what happens if this knowledge is insufficient? With the Internet, Google, Wikipedia, and YouTube we have "at hand" an infinity of educational resources. Their advantage is the immediate availability of resources; however, all of them belong to the "virtual" world, which in the language of physics, means possibly, but not necessarily, observable.

Therefore, we prefer real teaching, which aids both the experimental stands prepared in advance and ad hoc objects, such as balls taken from the pocket or cables of the computer used to draw an ellipsis on the blackboard or to show a reflection of a wave travelling in one dimension.

Taking into account the virtualisation of life (especially after the period of distance learning), it is important in creating knowledge and building correct interpersonal relationships to "touch" real exhibits. Real objects provide students with a full sensorial experience—with their weight, colour, and sound—when splashing. Real experiments yield not only expected experience but also a myriad of "failed" results. All these unsuccessful results become an opportunity to explain that in the real world (physical, chemical), many factors, not only those predicted, affect the result of the experiment. Additionally, according to the already cited "1:9" didactical principle, combined with Paul Newton's et al. [30] allowance for the teacher's fallibility, the real experience induces authentic educational joy [42].

Objects that can be found in everyday situations, table-top Newton's cradle, plasma lamps, and balancing toys (see our virtual collection [22]), are particularly attractive for students. They can be also easily adopted by teachers. A statement by pupils "I have already seen it" is the confirmation that the constructivist path can be applied. This holds

for all sciences; appeal to the "palpable" knowledge of students as a starting point and discuss their concepts about the subject. In teaching biology, Rebecca Garcia et al. [43] explained it as follows:

> We anticipated that this prior exposure would motivate students to apply existing information and expand on their perceived knowledge. In addition, we sought to dispel existing misconceptions through class discussion and evaluation of previously held concepts. [ ... ] Students are more receptive to learning science when they can directly relate the material to real-life situations and events.

Following this lead, our first recommendation is to base the didactics on real objects. As this happens in a time dominated by the virtual world, we call this approach neo-realism (N-R). The objects used in the lesson are familiar to students; usually, they are simple, small objects that do not distract them from the discussed laws and physical phenomena. For example, two similar-in-size balls, but of different masses (caoutchouc and ping-pong, or just two different nuts) are dropped from the same height, and they fall at (almost, but with the difference invisible) the same time. In this way, it is possible to show the independence of the gravitational acceleration from the mass, or to explain the exchange of energy between two falling balls as a result of a collision (see our multimedia clips [44]).

Then, in the constructivist scenario, we complicate the experiment. We divide a piece of paper into two identical halves and we make one of them fall as a crumpled ball and another as a kind of parachute; here we show how air resistance affects the fall of bodies. The lesson is completed either by an experiment with a coin and a feather falling in a glass tube without air inside and/or when do not have the vacuum pump, by the clip of Galileo's experiment performed by Apollo 15 astronauts on the Moon [45]. The didactical potential of everyday objects can be a starting point and encourage teachers to create their own scenarios.

Another example of N-R is a gravity funnel illustrating Kepler's three laws [22], and a kitchen funnel that illustrates deviations from these laws, in particular due to the General Relativity, see Figure 2.The conceptual analogies used in teaching, correlated with everyday objects, stimulate the imagination and cause the formation of new associations, thanks to which they are more easily absorbed by the listeners.

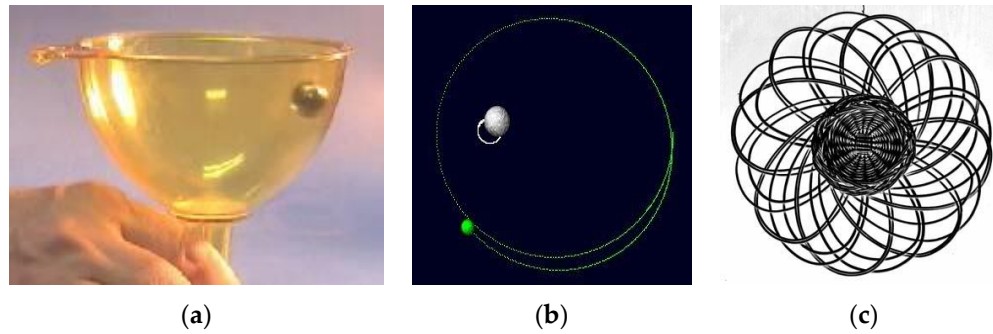

| (a) | (b) | (c) |

**Figure 2.** Neo-realism—any object is useful in explaining physics: (**a**) kitchen funnel as an illustration of general relativity; (**b**) open orbits for non-Newtonian potential, a simple computer simulation; (**c**) ethnographic artefact in wicker showing an open orbit of Mercury (collection GK). Reprinted from ref. [14] with permission, author GK.

Real objects can also prove themselves useful in the explanation of modern physics. Physical modelling can be used to explain guesswork in modern physics, e.g., Planck's blackbody modelling with a box with a small hole, mirror lasers with translucent sunglasses, and mass spectrometers with shaking containers of different-sized balls. Each model can be extended with more advanced experiments (measurement or computer-aided). The above examples of using simple, intangible objects are the idea of neo-realism. Their main goal is to illustrate often complex and abstract physical laws in the simplest possible way.

Together with objects, computer models, photos, descriptions, and video clips, it makes up the pedagogical idea of neorealism: everything that a student can touch or see should be touched, and even if objects are scientifically impossible to visualise, such as quarks or electrons, they should also be able to touch them (Figure 3). Unfortunately, practice often shows that teachers only occasionally use the benefits of the surrounding objects, what may be due to their insufficient didactic preparation and/or lack of imagination.

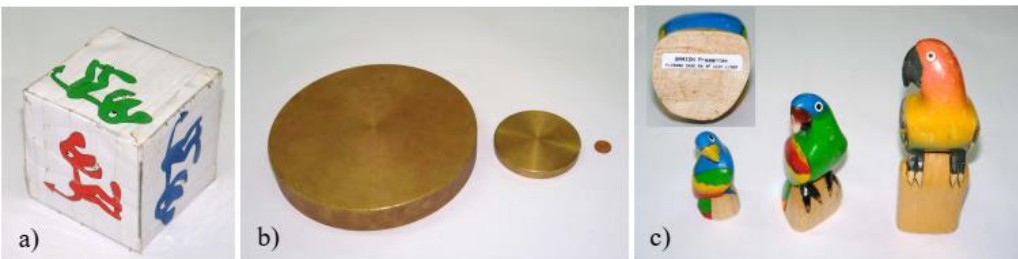

**Figure 3.** Neo-realism extrapolated beyond the limits of imagination. Typically, atoms, electrons, and neutrons are shown as small points, but similarly to Earth seen from space, also neutron (**a**), made of two quarks down and one up has its own structure; (**b**) assuming 1 euro-cent (Greek: lepton) as an electron, the mu and tau-leptons scale as bigger copper disks; (**c**) alternatively, we can imagine a neutron, proton, and the heavier hyperon lambda as three-colour parrots (collection GK, acquired in "Baryon" shop at San Paolo airport, 2005). Photos reproduced from ref. [14] with permission, (C) GK.

The use of experiments, obviously, is not new in teaching physics; it is the basis of many methodologies as stand-alone laboratories or as a part of the lesson in a class. For an example, see [46,47]. Additionally, demonstrations in the lecture hall have been proven to be a significant factor in triggering an interest in physics [48]. However, our experiments form a didactical sequence. It is often in our approach that such a sequence should reproduce the historical development of a physical concept. Let us say, the lesson on gravity starts from Aristotle's "teleological" statement, then we pass to the experiment, which is first qualitative, repeatable, but also abstracted from details ("close your eyes and listen to the ball bouncing, please!"), then to the measurement ("look and check, please"), then to the confuting of the simplified results ("now the two balls will fall from much higher altitude"), leaving the listener with heuristic anxiety. From Newton, we take the clear condemnation of any "spirituality": "There are some people who say that the objects may be moved just by thinking. They call it telekinesis. Let us try if it is real!". Further, following the warning of the Polish pedagogue, Kazimierz Sośnicki that too much exemplification leads to childness, we never repeat experiments unless with a clear scope to show additional didactical aspects.

In our practical applications, the two approaches, H-C and N-R, serve to obtain a cognitive goal that is not a mere knowledge of the notions of physics, but rather acquiring the ability to think. Students/pupils/teachers are invited to follow a line of reasoning, where at the beginning we do not reveal a clear point of arrival, such as "the law of gravity". Therefore, we use a variety of ways, not disregarding any a priori. So, using falling balls, we start from Aristotle's (wrong) statements, to make pupils arrive step-by-step but autonomously to the correct formulations. Further, we concentrate more on the complexity of phenomena, forming lessons on "the electricity sources" rather than on traditional subjects, such as electrostatics or electromagnetic induction. In this way, we avoid oversimplifications when students are not able to understand "a real nature"; they do not need this to remain fascinated by physics. In the following, we review some of our activities.

### 3. Teaching Environments

Taking our two general didactical principles, hyper-constructivism and neo-realism, we applied them in different environments, from primary school to university lessons, for active teachers.

*3.1. Primary School: States of Matter and Electricity*

We (KW) practiced H-C teaching first in primary school, within the official, national curriculum in the first grade (8 years old) by conducting lessons about electricity. However, we resigned from the typical organisation of the class, instead running a kind of workshop, in which children may touch (only if they are safe) all objects (Figure 4). The experiments for these lessons included everyday objects, such as plastic tubes, pieces of metals, lamps, batteries, etc.

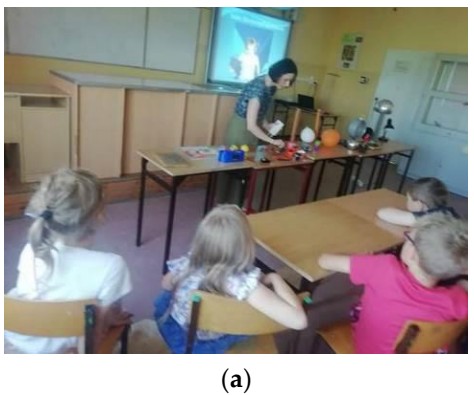
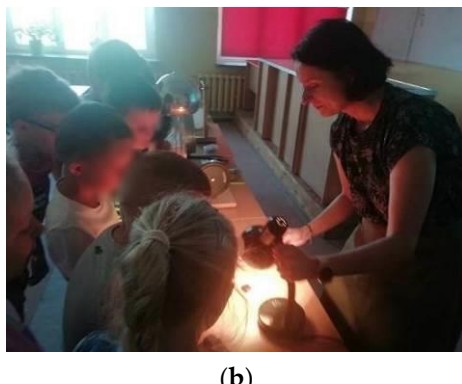

(**a**)                                                                                  (**b**)

**Figure 4.** An interactive lesson with students aged 8. (**a**) Teaching aids (traditional, interactive, multimedia) have been selected in such a way as to be able to refer to the knowledge that students at this age already have and those that helped to deepen this knowledge. We used batteries, fruits, a solar panel, a dynamo, balloons, straws, and many more; (**b**) working with solar cells in small toys. We show that a traditional lamp (i.e., producing also infrared radiation, and, therefore, resembling the solar spectrum) is more efficient than the "new", i.e., LED and fluorescent lamps with similar luminosity. Teacher: KW, (C) for photos: KW.

The lesson was easy and amusing, stimulated by ordinary objects that the pupils knew from everyday life. Together with the teacher, they were searching for answers to simple questions: "Can you comb your hair with a plastic ruler? What is it like? What would happen if . . . ? Can you separate pepper from salt?" (Cinderella's story). Properly thought-out experiments allow children to become acquainted with the research method from the early years of their education; it also plants a seed which can be used in later life.

It is clearly visible that teaching in the constructivist approach exposes itself primarily in the active attitude of the student, which is stimulated by the teacher using the appropriate aids. Then, the student is an independent and active subject who constructs his knowledge system, analyses, tracks his successes and failures, and draws conclusions from them. The teacher's role is mainly to create a situation of cooperation, stimulate the use of students' experiences, and facilitate the creation of new knowledge by asking questions. Therefore, language and vivid narration play an important role at the very beginning of learning.

Carefully planned experiments allow for conducting a lesson (Figure 4) on electricity (difficult in terms of didactics for the age of 8 yrs.) in a way that enables one to achieve the assumed didactic goal. Students are active participants in the lesson at each stage. They observe, experience, ask questions, and provide answers. Brainstorming between students leads to surprising results; the role of the teacher is merely to select correct conclusions, in accordance with the H-C didactical principles.

The achievements of the cognitive–constructivist theory point to the teaching methods that increase effectiveness. According to this approach, teaching is effective when the

student's activity is stimulated; he/she uses previously acquired knowledge, anticipates, draws conclusions, and independently formulates solutions. These activities take place when the student is the constructor of his knowledge in the mind. The pupils asked other teachers in the school: "When will we have lesson with Mrs. Kasia?", and this is the best evaluation that the pupils enjoyed our didactical approach.

The lesson on the states of matter was scheduled for the 5th form. At the beginning (Figure 5a), the pupils attempted to recognise presented liquids based on smell, colour, and viscosity. Then, they made a drawing (first attempt at making a hypothesis) of the supposed order the liquids would be arranged in when poured consecutively. Afterwards, they proceeded to the experiment. Students were invited to the teacher's table to pour the liquids. Next, they verify their hypothesis by comparing the result of the experiment with the previously made drawing. They experienced a huge surprise when the oil was a layer above the water. The outcome of the experiment (the oil layer on the top) came as a completely unpredicted solution.

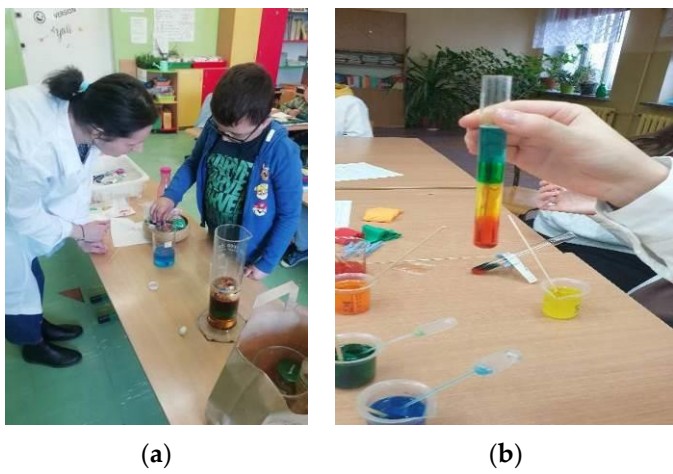

(**a**)  (**b**)

**Figure 5.** Fun with density: a group of 8-year-old pupils playing with Archimedes' law, making different fruits float in freshwater (**a**), then they check (**b**) that the buoyancy force depends on the density of the liquid (salty water, honey). Photo (C) KW.

Older students (Figure 5b) prepared four beakers with the same amount of water but with different amounts of sugar. The water had been coloured with water pigments (or, even better, with pigments that in some countries, such as Poland and Germany, are used for preparing Easter eggs). Then, they slowly poured it into the test tube, starting with the water with the most sugar. The students very quickly drew conclusions about what influences such an arrangement of layers. The more sugar there is, the higher the density will be. They found out why maple syrup is so thick. The aim of this sequence was to show that the division into (three) states of matter is somewhat artificial; the atomic structure is the key concept.

The main assumption of the lesson was to arouse the children's curiosity, which will strengthen their desire to explore the world of science and discover their own passions. The use of materials that children can find at home (different fruits, syrup, and colourants for Easter eggs) makes them "ambassadors" of further teaching at home.

### 3.2. Physics as a Playground

In elementary school, pupils already possess some cognitive tools to construct, with the help of the teacher, the correct notions: they read, they use smartphones, and they watch educational TV channels. A real challenge is to use the H-C methodology at the pre-school age, showing that the phenomena which children already observed follows some rules (that in further education, we will call the "laws of physics"). According to

Trumper [49], "a constructivist approach seen as a process in which pupils are actively involved in constructing the scientific concept should be presented as early as possible".

The same subject of electricity, that in elementary school we present with the commercial set of experiments (Volta's electrophorus, the electroscope, Wimshurst's electrostatic machine), see Figure 4, can be taught almost with "nothing", even merely a plastic tube and a piece of cloth (Figure 6a). Moreover, the same effect of rising hair can be detected due to the electrostatic charges one observes after sliding whilst wearing synthetic-fibre trousers on a long plastic slope (Figure 6b). What the (hyper-constructivist) teacher must do is spot the didactically interesting situation in the surrounding environment.

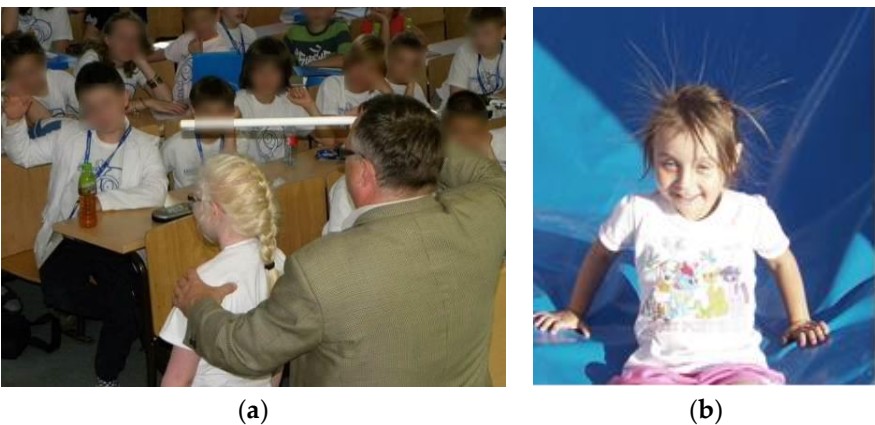

(**a**)  (**b**)

**Figure 6.** Showing electrostatics to children. (**a**) A plastic tube electrised by rubbing with a woollen pullover generates voltage up to a few keV. It is easier to demonstrate this on a girl with thin, blond hair. Photo Maria Karwasz; (**b**) Natalia Wyborska (age 4) and her first experience with static electricity in the playground: sliding in synthetic trousers on a long plastic "castle" generates electrical potential of tens of kV. Photo KW.

For children, experiments with movement are particularly attractive, for example, those involving balls, carts, and puppets. However, the very basis of the H-C lesson is a narrative story. One of our preferred (and also preferred by the public) lessons is entitled "Why do objects fall?". The standard answer, "Because of gravity", does not reveal much information; gravity is Earth's attraction and Earth's attraction is gravity. This is a perfect tautology; it is correct, but does not provide much information. So, we propose an alternative explanation dating to Aristotle: "objects fall because they are heavy, and the natural place of heavy objects is the centre of the Earth". So, we experiment whether objects fall to the centre of the Earth by placing a jacket (to avoid jumping) on a table, then on a chair (and not any further, to avoid sliding down to a mere phenomenology).

Narrative H-C teaching needs to be taught permanently in order to retain pupils' attention. So, the next immediate question is if the ball may jump up by itself. The answer is obviously not, unless we use a half-ball which if inverted (and posed on the table), jumps up, making it appear spontaneous [22]. So, Aristotle's explanation is not exhaustive enough. The keyword here is "energy". Children understand it perfectly; see Figure 7a. This picture is made by a 12-year-old girl, but the same story allows children aged 3–4 to equally enjoy the experiments; see Figure 7b. Note that we present a different way of arriving at the concept of energy from that which is practised in the literature, even if it is in a constructivist manner; compare [49,50]. Details of our scenario are provided in refs. [32,34].

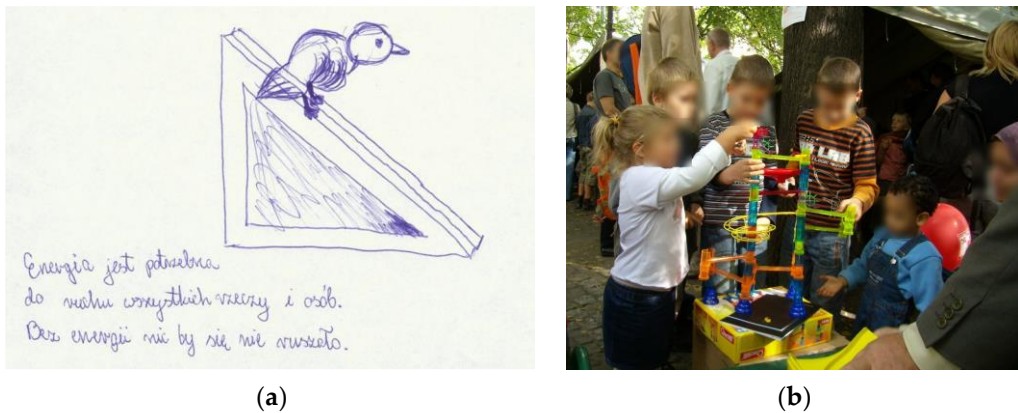

**Figure 7.** Towards the evaluation of the efficiency of H-C teaching. (**a**) Picture by a 12-year-old girl drawn six months after the interactive lesson "Why do objects fall?" The sentence says: "The energy moves all things". (Brzeg, Poland, March 2014); (**b**) pre-school children queuing to conduct experiments on balls sliding down (Warsaw, 2012). Lessons GK, photo Maria Karwasz.

*3.3. Interactive Lectures for Schools*

What is more demanding are lessons for the general public: it is difficult to keep attention of listeners if they have much diversified knowledge of science. We have practiced such lessons with constant success for around 15 years. In Figure 8, we present two implementations of chasing carts in two different scenarios: for a secondary school in the Republic of Korea and for elementary school children in Poland (at complementary teaching within a so-called children's university). The scenario was born from the observation by Galileo that stones of different masses fall with the same "velocity" (one should speak about the acceleration, but from the cognitive point of view, this would be an unnecessary complication). What makes the whole subject intriguing is the constructivist play; we pose the question of which cart, the heavier or lighter cart, is quicker.

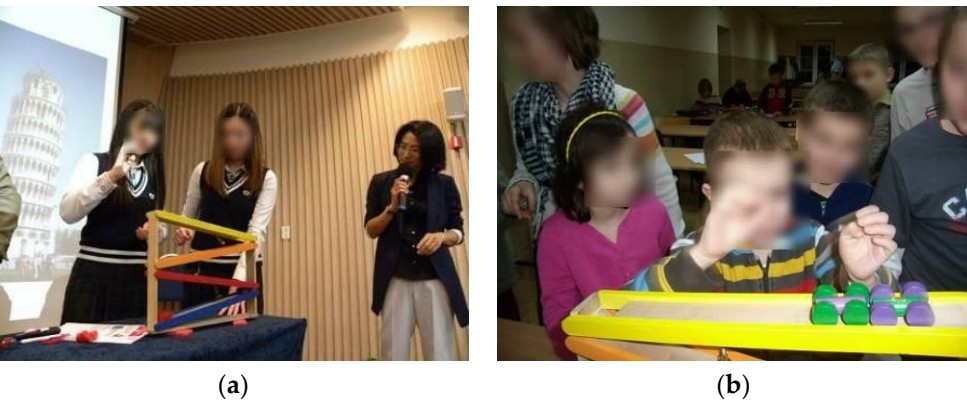

**Figure 8.** Galileo's experiment: which cart, heavier or lighter, is faster on the inclined plane? (**a**) Preparing the scenario. Lesson by GK as a public lecture in Gunsan (Republic of Korea, 2016); (**b**) autonomous trials, university for kids, Gorzów, Poland 2012. Photos by Maria Karwasz.

First, a pupil with closed eyes identifies which one is heavier. Then, the audience votes, similarly to in a parliament, on the two alternative working hypotheses. Only at that point the scenario (Figure 8a), "Ann starts with the lighter cart. When it arrives at the end of the first slope, Betty launches the heavier one. If the heavier is quicker, it should reach the lighter by the end of the last slope, should not it?", "No? Probably we made an error. Let us try again!", "So, let us check the alternative hypothesis". The whole series of 2 + 1 launches lasts for little more than a minute, but it triggers such an interest that after the

lesson, everybody, regardless of their age, "wants to try it on his/her own, please!"; see Figure 8b.

### 3.4. Working with Teachers

Surprisingly, working with teachers is not easier than working with children. Teachers are stubborn about their methodologies. The OECD TALIS study [11] showed that, in particular, Italians prefer traditional teaching (based on textbooks) to constructivist approaches. Poles, in the same ranking, are in the middle, between Iceland, Austria, and Australia on one (constructivist) side, and Bulgaria, Malaysia, and Italy on the other.

However, practising the H-C approach in Italy, we noticed that being accustomed to traditional teaching makes Italian teachers much more perceptive than their colleagues in Poland. As seen in Figure 9, they react with enthusiasm to the novel methodologies, both in the form of interactive lectures as well as free-hand workshops. Teachers are very creative, and their long-term involvement in the innovative didactics remains high, even several years after lessons, as we observed from their constant scientific activity [19,20].

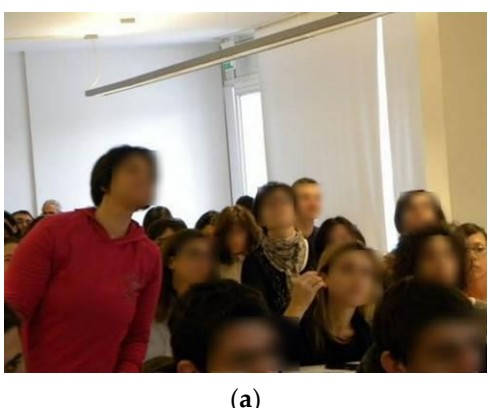 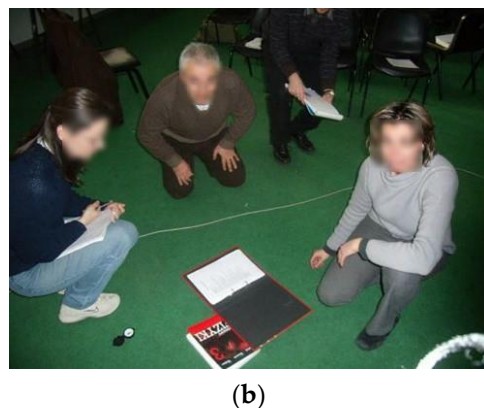

(**a**)            (**b**)

**Figure 9.** Triggering heuristic interest among teachers: (**a**) "Why do objects fall?"—an interactive lecture for secondary-school teachers at the University of Macerata, Italy, 2014; (**b**) free-hand experiments with magnets for teachers at the University of Udine, Italy, 2009. Lessons GK, photo Maria Karwasz.

This interest is confirmed by continuous invitations from different didactical entities (primary and secondary schools, universities) all over Italy (Milano, Bolzano, Udine, Trento, Cagliari, Reggio Emilia, etc.). What is appreciated by teachers (in Italy, and in general in different countries, as we observed in GIREP congresses) is the interdisciplinary character of our approach: mixing portable, interactive experiments with concepts deriving from the history and philosophy of science. To quote some opinions (from the 2022/23 courses for Italian teachers), "Thank you Professore for your precious advices" (Giuseppina), "The work by the teacher is like a sculptor" (Luigina), "Thanks a thousand for the opportunity to participate" (Alice), "Thank you for having divided with us your immense culture" (Maria), and "Lessons wonderful, never boring, reach with many interesting reflections" (Francesca).

In Poland, we have developed a constant forum of contacts with teachers via a yearly organised seminar (held in December, at the name-day of Copernicus). This initiative is divided equally into current progresses in science (physics, astronomy, biology, geography) with lectures held by active scientists, and workshops run by teachers. The seminar gathers each year around 50–70 participants, many of whom are returning participants. It became, with its 15th edition in 2022, not only the channel of scientific and methodological updates for teachers and researchers, but also a forum for social engagement.

### 3.5. Lessons for Wide Public

A real challenge for presenting physics (but on the other hand, an opportunity) is conducting meetings with the general public that include not only pupils, students, their

parents, and teachers, but also persons from local administration, authorities, and enthusiasts of science. We practice such events mainly via science festivals, which are organised by local authorities in collaboration with a school that assure logistics. The members of the public in such meetings are vast; they make up a few hundred people. It is challenging to prepare different subjects every year that are also interesting to people with various interests. Such general subjects are, for example, "An industrious PhD (she) student" (about Marie Curie), "It sounds good"—about sounds and different, strange instruments, "Far, far away", a study on the different landscapes on Earth, as well as subjects such as the practically of international scientific collaborations, and so on. These initiatives last for years, proving that the educational messages transmitted and the ways we present them receive a long-standing acceptance. Note that the best place for meeting the general public are small- and middle-sized towns that are rather peripheral from big cities.

Conducting lessons for the general public is tiring, but a fast and fluent narration interlaced properly with experiments allows also for such an environment to trigger interest, and at the end, also involve the audience; see Figure 10b. Obviously, scenarios must be within reach and must offer non-trivial experiments. The members of the public amount up to several hundreds of people at such lectures; a measure of the lectures' success is not only the attention attracted but also the (frequent) wish of teenagers to take a photo with the professor.

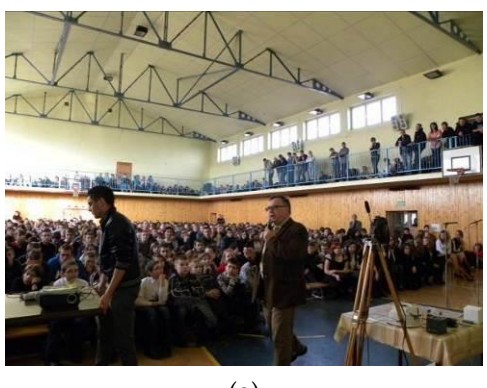 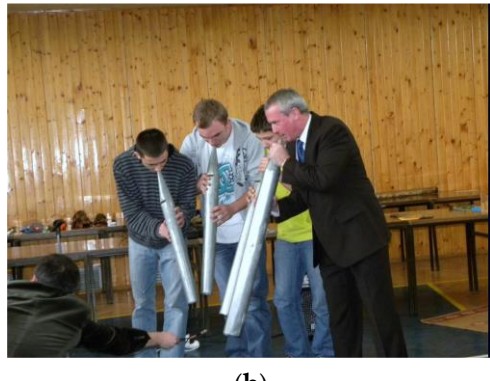

(**a**)  (**b**)

**Figure 10.** Lessons for wide public—Regional Science Festival in Nadroz, Poland. (**a**) A 2011 lecture with experiments on modern physics; (**b**) 2010 "It sounds good!", an interactive lecture on acoustics, sound, and music: the director of the school playing organ tubes with his students. Lecture GK, photos Maria Karwasz.

### 3.6. Projecting Scientific Exhibitions

Science centres date back to 1960s ("Exploratorium" in San Francisco created by Oppenheimer). In Poland, the very idea was absent until around twenty years ago. In 1998, we (GK) organised, in collaboration with Trento University (Italy), the first exhibition of small interactive objects, "Physics and Toys" [22], which was presented at the national congress of the Polish Physical Society. This boosted the demand for such an exhibition, and also for science centres.

In 2009, the "Hevelianum" (named after the XVII century astronomer) Science Centre was created in Gdańsk and, a year later, the "Kopernik" Centre was founded in Warsaw. The former encompasses around 250,000 visitors each year, and the latter about a million. These numbers are close to the population of the two cities; the visitors are mainly scholars from the centres' surroundings. Physics is the core theme of these centres. In the "Hevelianum Centre", the same elements that were interactive objects at the "Galileo inclined plane" didactical tunnel are used with different arrangements; this is an example of more fun, less direct didactics (Figure 11a).

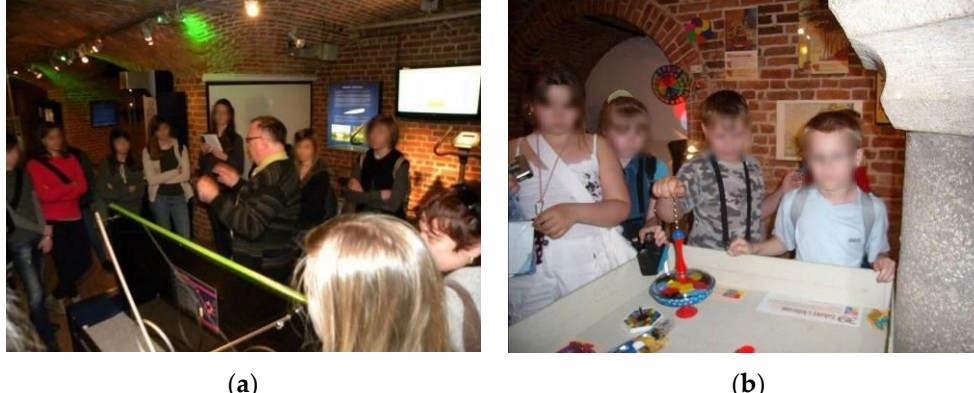

(**a**)                                              (**b**)

**Figure 11.** (**a**) Ad hoc hyper-constructivist lesson at the Hevelianum Science Centre, Gdańsk, for the secondary (pedagogical) school from Trento, Italy: a series of interactive experiments performed with the "death-loop" (at the bottom of the photo) allows to explain the concepts of the centrifugal force, of the potential and kinetic energy, the independence of the mathematical solution from the mass of objects but dependence on their inertia moments. An object is only an "excuse" for constructivist teaching; (**b**) "Fiat Lux—playing with light", an interactive exhibition, with 20 editions all over Poland (2007–2012): colours, geometrical optics, interference, diffraction, arts, history of science, etc. Grouping of objects into "nests" and very short explanations make different subjects self-explanatory. Authors: GK and Michał Kłosiński (Museum Toruń). Photos: Maria Karwasz.

We conducted experiments in different forms of interactive exhibitions:

- On movement and energy at "Hevelianum Center" (2009), as previously mentioned;
- An interactive exhibition on physics and the history of optics "Fiat Lux—playing with light. From Vitelo to optical tomograph" with 20 editions all over Poland and 100,000 visitors (2009–2012);
- Exhibitions on sounds (at the national congress of the Polish Physical Society in 2003, and a remake in 2023), and so on.

These exhibitions are not mere accumulations of objects; there is always a (hidden, hyper-constructivist) idea behind them. For the "Going downhill" experiment, the full tittle was "Everything on the inclined plane of Galileo, in other words, how the potential energy converts to the kinetic one, and how one may have fun with it". The sequential series of inclined planes with object rolling, falling, and sliding down served to introduce step-by-step notions on kinematics and dynamics. The exhibition on optics was multi-faced; it included the topics of physics, history, and arts of optics (Vitelo, Kepler, Newton, up to impressionists and Picasso, etc.).

We monitored the reaction of the public by the "book of visitors". Generally, the notes in this book were enthusiastic: "I am in the 5th [elementary] class and I think that the exhibition is great. I liked the most that one could touch everything.", "After having visited the upper part with paintings and arts, my son was pleased but tired. When we went down the world of [optical] illusion he revived and was astonished with all these novelties." (Mother of Max 11 yrs. old.), and "I live in Ireland—here is cool!" (Olivia). The shortest comment was by made by a beggar who came upon the exhibition by accident: "Sir! How it is interesting. Eyes turning around!". In view of frequently heard opinions that physics is unattractive and difficult, this was a great success, considering also that we are not professionals of scientific exhibitions.

The point is that when the ideas behind a given exhibition are known, it is easy to construct lessons ad hoc. Say, the exhibition on optics allowed us to run narrations (illustrated by experiments) on geometrical (Newton's) optics, on the diffraction, on the emission spectra in atomic physics, and so on, depending on the inventiveness of the teacher. A "visible" outcome of ad hoc lessons at science centres, see Figure 11a, is the increased interest of students, seen not only in their reactions but also in a durable change

in their attitudes towards science, seen by students of the pedagogical lyceum choosing the profession of teachers of mathematics or the carrier of the theoretical nuclear physicists at CERN. More details on our scientific exhibitions are provided in ref. [14].

### 3.7. Constructivistic Text-Books

The idea of presenting physics by "touchable" objects and concentrating on the main features and not on the details was applied to a series of our books, which include "*Astronomy for kids*", "*Mechanics*" [51] for the first year of lower secondary school, and "*Modern physics*" for the first year of higher secondary school. We entitle these works as "tex-books", to underline that they are not traditional textbooks, but rather "handy books" (play on words in Polish). A detailed study of the didactical efficacy was performed (by M. Sadowska and KW) for "*Mechanics*" [52]. Two schools in a middle-sized town (Kalisz) and a village (Dąbrowa Biskupia), respectively, comprised together about 200 students.

The "Toruń physics tex-book. *Mechanics*", which appeared on the market in 2010, was primarily aimed at increasing students' interest in physics and astronomy, but it also included elements of the structure of matter and chemistry (Figure 12). The book provides information "from the scratch", starting from defining systems of reference (for example, the radial system for a spider in its net) and the rules for the evaluation of approximated numbers in mathematics, physics, the economy, and pharmacy.

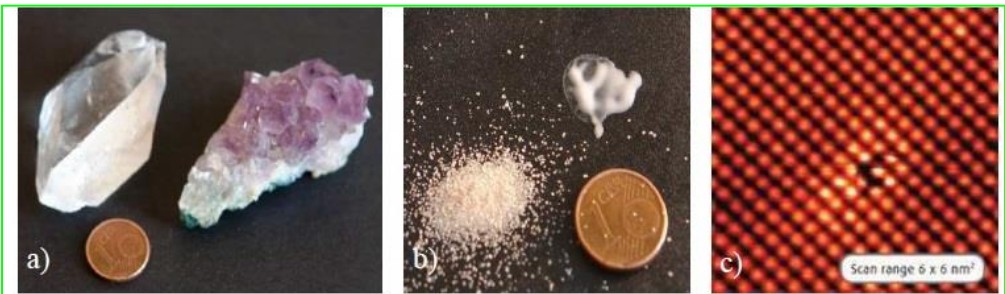

**Figure 12.** Hyper-constructivist tex-book (not "textbook") [51]: touching the dimensions of an atom, by dividing the crystal of quartz (**a**); into sand by one thousand times (**b**), and again by one thousand times ((**b**), upper right corner) into a polishing liquid. Photo (**c**): picture of silicon crystal seen by atomic force microscope, 6x6 nm$^2$, Omicron. Text and photo GK. Reproduced from ref. [51] with permission.

What distinguishes our tex-book from other textbooks available on the market is the way of narration. The questions asked in the text help the student to explain the laws of physics by combining common knowledge with scientific knowledge. We introduce students to the elements of mathematics that are essential in the study of physics. Authors cite the example of the European Union budget, from which the student learns for what purpose approximations should be used and learn the rules governing rounding numbers. It is also worth noting that the concepts new to the students are set in real, practical contexts. An interdisciplinary approach is used, making reference to other fields of science and everyday phenomena, to some extent, in a similar way as done by Paul Hewitt in Conceptual Physics.

When introducing new notions, we select the most important, avoiding the information which is not essential for further narration. For example, we do not start from the traditional, Empedocles-like states of matter, but we rather introduce first to the concept of atoms, see Figure 12, and only then do we distinguish solids, liquids, gases, and plasma as different organisations of atoms.

Obviously, our tex-books do not start from "null". The very basis is the excellent five-volume "*Physics for Inquiring Minds*" [53] by E.M. Rogers, dating back to 1960, that considers about physics, history, and philosophy; the narrative, illustrative, and modelled

notions on kinematics were inspired by the interactive CD by the Italian nuclear physicist Ugo Amaldi [54].

We tested two subjects from "*Mechanics*" [51]: (i) energy, work, and power and (ii) kinematics and compared the results with another Polish textbook [55] of a well-established publisher, officially approved for national-wide teaching. Questions were divided into three categories: knowledge, understanding, and abilities to use in typical and problematic situations. The synthetic results are shown in Figure 13. In the experimental group, the percentage of positive results of the test was 81% (and the mean score was 2.31 out of 5.0, which was the maximum), it was only 56% (1.75) using the traditional textbook. A particularly high rise in efficiency was obtained in the knowledge and use of knowledge in problematic situations (an almost 50% rise in positive answers). An even better result, almost 100% positive answers (and a mean score of 3.54) were obtained for the subject of kinematics (the mean score in the control group was 2.43); Figure 13b.

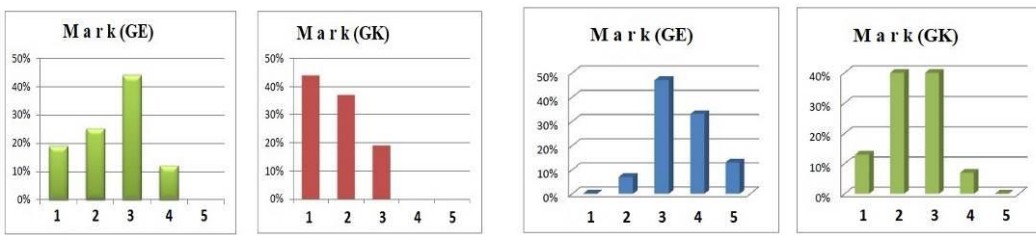

**Figure 13.** Comparison of experimental (GE) group using "Toruń tex-book" with a control group (GK) using a traditional textbook: the scores in GE groups are significantly higher than in GK, and with much uniform distribution. Left panel—test on energy; right panel—kinematics [52].

The most significant effects on the learning outcomes between our textbook and the standard one were noted in problematic situations, where the results in understanding and applying knowledge were included. While the experimental group of the program had an average score of 57%, the control group obtained 39%.

In the subject of energy, our tex-book assured a better understanding of the very concept, apart from a classification of the types of energy (kinematic, potential, etc.), as there is in traditional teaching. Faced with the alternative between the H-C tex-book and the traditional one, 94% of the students preferred the former for work within school (and 72% for homework). The tex-book is useful for reaffirming students' understanding (83%), repeating notions before an exam (94%), obtaining interesting information (94%), and learning numerical exercises (89%).

## 4. Discussion

The H-C method develops scientific thinking based on asking questions, planning the research process, and drawing conclusions from the conducted experiments. The goal is the learning process itself rather than the final concept; in the lesson on falling objects, we arrive at the concept of "energy" rather than the statement that "objects fall due to gravity". This is the first difficulty in the quantitative comparison of H-C with traditional teaching.

What differentiates our scenarios from simple "hands-on experiments" or the SPEE (situation–prevision–experiment–explanation) is the grade of interactivity and students' creativity. Depending on the teaching environment, for the same subject, we can dose differently the surprise, the unknown, the predictable, etc. In the typical SPEE situation, say on electromagnetic induction, see Figure 14a, one presents an experimental set-up: a coil, a magnet, a current meter, and wires. It is clear that wires should connect the coil with the meter, and that the magnet should be inserted into the coil; there is little to be invented. Therefore, traditional textbooks concentrate on the very formal question: "Which is the direction of the electric current induced?". This requires very little imagination or real discovery, making physics boring.

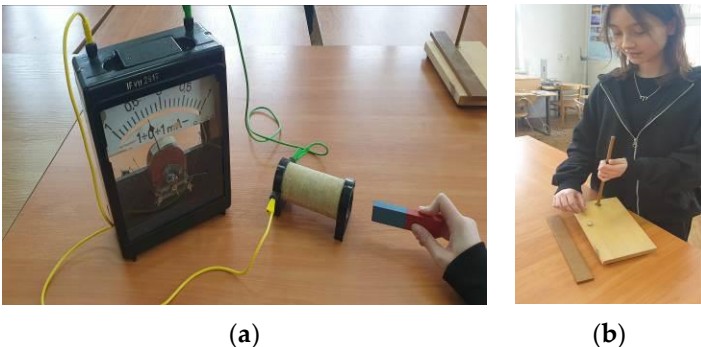

|(**a**)|(**b**)|

**Figure 14.** Traditional and H-C experiments on electromagnetic induction. (**a**) Introducing a magnet into a coil creates an electric current; (**b**) magnets, in general, interact with external magnetic fields (similarly to Earth's field, when rolling down on a wooden plate) and with electric currents (for example, induced in a copper bar when sliding on it). Photo KW, featuring Ewa Wyborska.

Instead, in the subject of electromagnetic induction, one can propose a whole series of simple experiments of magnets falling inside copper tubes or sliding (with a constant velocity) on a thick copper bar; see Figure 14b. Recently, we have applied such setups to modernise the rather traditional teaching for students in engineering at Kazakh National University. Simple experiments, such as that depicted in Figure 14b, were augmented with virtual means, i.e., internet descriptions and video clips. Additionally, in that environment, we observed a significant rise in interest and of the didactical outcome: all students expressed an opinion on the usefulness of such blended methods, and knowledge tests showed a rise in the correct answers, by 100% in some questions; while with traditional methods, this rise was modest (20%) (see ref. [56]).

Our hyper-constructivist, experiment-based teaching should be compared with two similar, but somewhat alternative, methodologies: the Socratic heuristic (maieutic) method, as invented by Socrates, and narrative, metaphor teaching. The Socratic method is still a masterpiece of arguing and debating, to the same extent in mathematics and in law [57,58], but physics is a practical, everyday science, now with centuries of experiments behind it. Using the N-R methodology, we need not to base only on the pupils' mere reasoning, but in any moment of didactical difficulty, we can recall an "ocular" experience: an experiment "from my pocket", or the experience already seen by pupils, or a video clip from the internet (such as objects falling in a huge vacuum chamber).

Second, our interactive, constructivist teaching is more than merely narrative; these approaches are complementary. The narrative teaching (of physics) in early childhood stimulates the imagination [59,60], probably to a greater extent than our experiments, but the free-hand reports of pupils some months after our lessons, such as those depicted in Figure 7a, prove that the H-C teaching leaves a durable "imprinting".

As already mentioned, the standard evaluation of the didactical efficiency of the H-C approach is a hard task. If we declare at the beginning of the interactive adventure on discovering the concept of energy "Today we will define the potential and kinetic energy", the whole possible heuristic path disappears before we have even started it. Additionally, in traditional teaching, we should declare that "Today we will speak about gravity, and we will also learn about the potential and kinetic energy". The two ways of teaching are to some extent incompatible. The H-C approach is more fascinating, but the formal aspects of the knowledge may be compromised; we are aware of this difficulty. For example, we do not explain that the electrical current is a flow of electrons [61] (that is not the full truth, as in semiconductors we speak also about "holes" and in liquids and plasma, about ions). Without such (simplified) notions, our pupils would probably be less successful in school tests, but they will certainly be much more creative than a "control" group. In a similar way, we do not fight the so-called misconceptions of pupils [62]; instead, we use such preliminary knowledge as a starting point for the H-C path. In our teaching, there are no wrong answers.

A clear indicator of the didactical success is the continuous request for our lectures, both in Poland and abroad. In Poland, in the last 12 years, the UMK team (5 researchers in total), using the H-C methodology, concluded approximately 300 interactive lectures, lessons, and workshops; see Figure 15a. Additionally, a real reward for the lecturer is the interest of the audience after the lesson, regardless of whether they are preschool pupils, see Figure 15b, or university researchers.

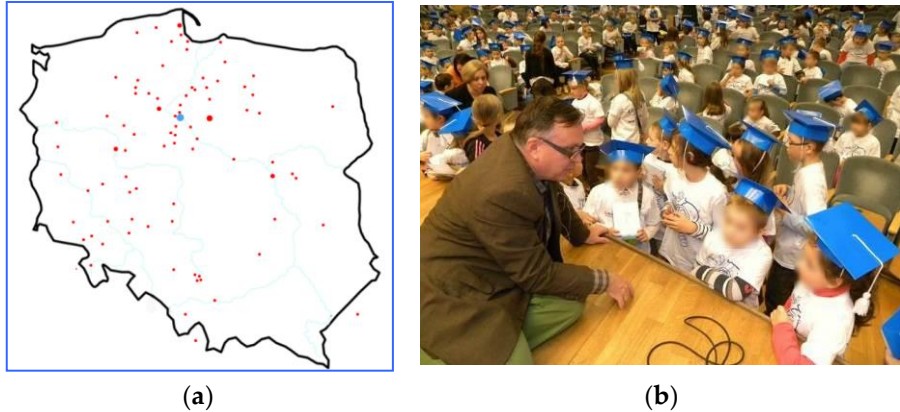

(**a**)                    (**b**)

**Figure 15.** Evaluating the H-C teaching of physics in different environments: (**a**) The geographic distribution of lessons in the period 2009–2019. The size of the points is proportional to the number of events. Source: own work, database http://dydaktyka.fizyka.umk.pl/nowa_strona/?q=node/523 (accessed on 8 January 2023; (**b**) the deep emotional involvement of young students after the lesson on electricity. Zielona Góra, October 2010, Source: own work, photo Maria Karwasz.

However, the real success is not in the mere number of lessons, but rather the flexibility of the "Physics is Fun" methodology to involve an audience of all ages, and also to induce social competencies, such as group collaboration, see Figure 16a, or practical abilities, similarly to written reporting, as in Figure 16b. The narrative approach also allows us to make teaching inclusive [23]; see, for ex., in photo 8b, the child autonomously performing "Galileo's" experiment by rolling down carts.

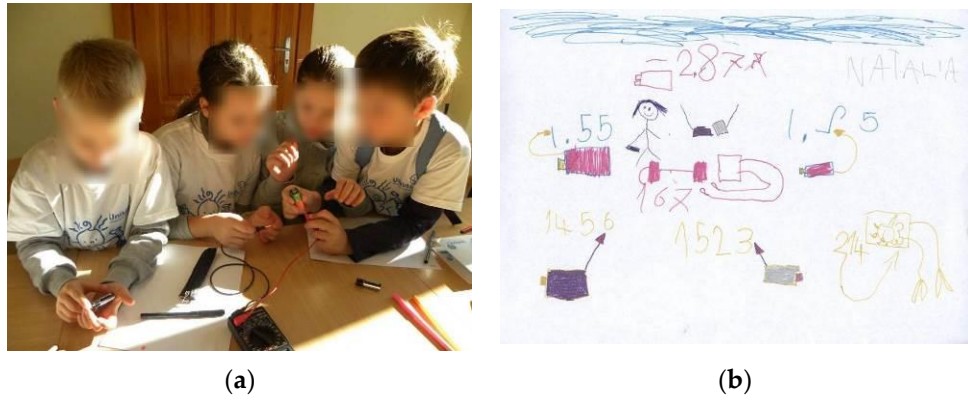

(**a**)                    (**b**)

**Figure 16.** (**a**) Pedagogical spin-off results of H-C teaching: autonomous division of tasks in measurements with a voltmeter: girls (10 yrs. old) plan, boys perform measurements and write the report. Głogów, Poland, 2014; (**b**) Natalia (7 yrs.) still hardly writes (note the inverted "2" number), but measured (and reported) correctly (even if without the decimal point) the voltage from new and exhausted piles and a vegetable (Al/Cu) battery (right lower angle). Świdnica, Poland, 2013. Workshops by GK, photo Maria Karwasz.

Compared with similar activities, such as open lectures organised by other Polish universities, in lectures and workshops at Science Centres ("Kopernik", "Hevelianum")

and interactive theatres ("Physics on Stage" at CERN [63], "Mysteries of Light" by Marco Giliberti in Milan [64], etc.), we observe a constant high degree of interest for our "Physics is Fun" activities. Notably, we run them without specific external support and the activity is continuous; practically all of the windows available in the great lecture hall at the Faculty are filled with our lessons, even if dissemination is not our main institutional task.

## 5. Conclusions

The present paper, initially conceived as a presentation of teaching methodologies, developed into a review of these methods and their applications in various didactical environments. Both neo-realism and hyper-constructivism methods began as our practical activity; the former started as university didactics, under the Bologna process in Trento (1993–1994), then developed into interactive exhibitions in Italy (1994) and in Poland (1998), which subsequently led, in a few years, to the formation of science centres (in Gdańsk in 2009 and in Warsaw in 2010). The was H-C developed mainly as a teaching method at children's universities (starting from 2009) and in interactive exhibitions (from 2008). Now, thanks to "tex-books", we have expanded the N-R and H-C didactics to school curricula. This continuous growth of activities testifies the success of both approaches.

In the period of September–December 2022, we delivered interactive lectures on university premises, under the title "Energy, part II Thermodynamic", for approximately 1000 students of primary and secondary schools; lectures and workshops on astronomy in Poznań for 100 children within a university for children; and lectures for around 300 students at the Science Festival in Kwidzyń, amongst others. The invitations for collaboration (a series of lessons for teachers on interdisciplinary didactics) arrived in 2022 from Italy and Kazakhstan, apart from the already mentioned contacts with the Republic of Korea. Work towards this is in progress.

We continue publishing H-C books. A book on science and philosophy [65] was published in 2019 in Rome and is now available on all major Internet vendor platforms, "*Modern Physics*" was re-published in 2021 by our University Editor, and "*Astronomy for Kids*" [66] arrived in 2023 as a reprint of the second edition. This proves the need for a somewhat humanistic approach to sciences, contrary to what had been shown by great masters in the past [2–5].

On the other hand, we have observed convergence with recent tendencies in science divulgation at the European level. A Spanish editor launched a series of multi-disciplinary and multi-sectorial (science, arts, engineering) books, comprising mathematics [67] physics, cosmology [68], and philosophy. Additionally, these series use simple language, a narrative approach, and numerous transversal references, but at the same time are based on the most recent scientific research.

New approaches to the didactics of physics, based on simple, possibly interactive experiments and an interrogative narration with the public, assured the success of a didactical approach in all different environments, both in schools and extracurricular teaching. This success is based on vast resources of both objects and scenarios. However, the hyper-constructivist implementations require the didactical capacities of teachers, lecturers, technicians, etc., to be advanced. Therefore, the long-term training of teachers is essential. The success of our recent (2022/2023) online lessons for Italian teachers of primary and secondary schools, on i) interdisciplinary, ii) multimedia, and iii) inclusive didactics proves this point.

In conclusion, physics needs not be boring or unpopular, but we must change the ways of teaching the subject to achieve this, particularly by adapting to specific learning environments whilst maintaining the essential characteristics of physics as a science which is experimental (Galileo), rational (Newton), and philosophical simultaneously. Teaching must involve learners in a constructivist way and respect their pedagogical needs. These conditions are hard to fulfil, but, in our opinion, necessary to keep the teaching of physics not only alive, but also fun.

**Author Contributions:** Conceptualization G.P.K. and K.W.; methodology, G.P.K. and K.W.; validation K.W.; investigation, G.P.K. and K.W.; data curation, G.P.K., writing – original draft preparation, G.P.K. and K.W., writing – review and editing, G.P.K. and K.W., funding acquisition, G.P.K. All authors have read and agreed to the published version of the manuscript.

**Funding:** This research has been co-funded by the Faculty of Physics, Astronomy and Applied Informatics, University Nicolaus Copernicusnstitutional.

**Institutional Review Board Statement:** The study was conducted according to guidelines of the Declaration of Helsinki, and approved by the Ethics Committee of General Data Protection Regulation (protocol code EU 2016/679 and 27/04/16).

**Informed Consent Statement:** Informed consent was obtained from all subjects involved in the study.

**Data Availability Statement:** Additional data on this study can be obtained at internet page: http://dydaktyka.fizyka.umk.pl/nowa_strona/?q=node/1009. (accessed on 2 February 2023).

**Conflicts of Interest:** The authors declare no conflict of interest.

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
