# Peer review of "How Constructivist Environment Changes Perception of Learning: Physics Is Fun"

_education, doi:10.3390/educsci13020195_

Round 1

Reviewer 1 Report

Review Report

Manuscript ID: education-2184542

Title: How constructivistic environment changes perception of learning: Physics is fun

It is no doubt that Physics is among the hardest subjects, together with Math and Chemistry. The authors of this manuscript show a careful approach to identifying problems regarding the motivation of students to accept knowledge covered by Physics and introduce some novel strategies, hyper-constructivism. It was a pleasure reading this manuscript. This study has the potential to be cited. 

I strongly recommend that the Editorial office consider this manuscript for publication after minor revision.

Reviewer’s Suggestion

Lines 31-31 "A opinion of the taxi driver at San Paolo (Brazil) airport reflects it concisely: “It was not my favorite subject at school”."

I would suggest to authors to omit this statement. The opinion of a taxi driver is not meritorious. He chose not to go for higher education, even in the scientific area which excludes or requires minimal knowledge of Physics. It cannot be concluded that his choice to become a taxi driver was solely under the influence of Physics.

English expressions must be used in Figure 1, not just in figure captions.

Lines 65-67 "The reasons are three-fold. The first is the very structure of physics as a science, which requires necessarily both the theory and the experiment. This is not the case, say, of philosophy or biology." Are the authors sure that the experiments in biology are not necessary? This has to be rephrased.  

Lines 133-134 "Lee Shulman is to be considered the third founder of the innovative didactics. He introduced the concept of the Pedagogical Contents Knowledge [10], ….." Here the authors can introduce the abbreviation PCK. This abbreviation is constantly used further in the text.

Lines 177-178 "The starting point in our methodology is critical thinking, i.e. Cartesian’s basis to modern scientific methodology, used also by OECD in AHELO evaluation of university systems …." Define acronyms OECD and AHELO.

Figure 12. Authors have to cut the text in Polish just under the pictures presented in this figure.

Figure 13. As I understand the word "oceny" is the Polish word for the word "mark" in English. 

Author Response

Thank you for the comments. All suggestions have been included.

Reviewer 2 Report

Review Report

Article title: How constructivistic environment changes perception of learning: Physics is fun

The manuscript deals with a current topic, which is the change in the way physics is taught in different types of schools and in extracurricular environments. The authors rely on two didactic principles in their activities: hyper-constructivism and neorealism. They give examples of teaching some physics topics in different settings: from school class to workshops for 3-4 years old children, interactive lectures for Kids Universities, ad-hoc explanations in science museums for secondary school students, up to public lectures in didactics at international congresses. In this context, I greatly appreciate all activities of the authors, which should lead pupils, students, and the general public to the opinion "Physics is fun".

Specific comments:

1. This manuscript does not have a character of an original research. It does not report scientifically based experiments. Research manuscripts should comprise Introduction, Materials and Methods, Results, Discussion, and Conclusions. The submitted manuscript does not follow this structure. Full experimental details must be provided so that the results can be reproduced (see instructions for authors available at https://www.mdpi.com/journal/education/instructions).

2.  The paper is closer to a review type of article. The authors must change the article structure to review.

3.  The title should be reformulated in the following manner: How a constructivist learning environment changes the perception of learning: Physics is fun

4. The introduction should include the objective of the study. Furthermore, it is necessary to provide a state-of-the-art review of current scholarly literature on different approaches to physics teaching and include it in the paper.

5.  The research questions and research hypotheses being tested need to be clearly mentioned at the end of the “Introduction” section.

6.  The section “Discussion” is completely missing. The authors should discuss the results and how they can be interpreted in terms of the research hypotheses. Have the results been compared with the literature? Have you found any similarities or discrepancies with previously published data? What are the benefits of the study for the future?

7.  All acronyms, such as PCK, H-C, UK, UMK, EU etc., should be defined in the text as follows: the Pedagogical Contents Knowledge (PCK), hyper-constructivism (H-C), etc.,

8. Do the authors have a written permission to re-use the previously published figures (Fig. 1, 12, 14, 15)?

9. Several figures are not properly formatted. Figures  a/ b should be cropped to have identical dimensions (Fig. 2 a) b) c) Fig. 4 a) b); Fig. 5 a) b); Fig. 6 a) b); Fig. 7 a) b); Fig. 9 a) b); Fig. 10 a) b); Fig. 11 a) b); Fig. 14 a) b); Fig. 15 a) b); Fig. 16 a) b). Furthermore, all figure legends should be translated to English.

10. The authors claim: ”The third, most important reason for poor performance of students in Physics (particularly in Poland) is the traditional,  transmission-like way of teaching, with few experiments, almost no multimedia and lacking any interdisciplinary connection.“ It must be supported by the results of scientific research.

11. The graphs in Fig. 13 should be marked by a), b) have clear legends “a) – test on energy, b) – kinematics“. Labels on x axis (Fig. 13) must be explained in the text. The authors used 3 categories “Questions were divided into 3 categories: knowledge, understanding, abilities to use in typical and problematic situations.“ Therefore, the results should be presented according to the mentioned categories. The results in Fig. 13 must be discussed. The authors state that they tested two subjects “Energy, work, power”, and “Kinematics” using “ToruÅ„ hand-book” and another textbook of a well-established publisher, officially approved for national-wide teaching. A reference must be provided.

12. All humans in the photographs must be blind-folded to assure the anonymity. Otherwise, a written consent of the person depicted is required.

13. The section “Summary“ needs to be better unfolded, highlighting what is new in the research and how the outcomes of the study can be used in practice.

Author Response

Thank you for the comments. All suggestions have been included. English extensive check by the native speaker has been done. The paper has been re-organized towards a review, as suggested. Extensive references added. 

Round 2

Reviewer 2 Report

Dear Authors,

I appreciate the great efforts you have made in response to my previous questions and comments. The revision clarifies almost all points I raised. You have significantly improved the clarity of your writing and addressed most of my concerns. Since you changed the article type to Review, it is necessary to change it also in line 1. A chapter titled Discussion is still missing in the manuscript. You have to either include it or use a combined Results and discussion section. The text of the manuscript needs to carefully proofread to avoid misprints, e.g., line 141 Piergiorgio Odifreddi (mathematician) e Antonino Zicchichi - Piergiorgio Odifreddi (mathematician) Antonino Zichichi, lines 337-338 summarizedit as follows - summarized as follows, etc.

 Kind regards,

The Reviewer

Author Response

We have changed the character of the paper into "review" - adding vast comparisons and enlarging the list of references. In the view of this change, the chapter that initially was conceived as "Evaluation of the impact" has been changed into "Discussion". We hope that this solution is acceptable. Some minor changes (misprints and repeated wording like "teaching. Teachers") have been done. We thank you the reviewer for all observations